# The critical factor: The role of quality in the performance of supported accommodation services for complex mental illness in England

Nerea Almeda[1]*, Carlos Ramón García-Alonso[2], Helen Killaspy[3], Mencía R. Gutiérrez-Colosía[1], Luis Salvador-Carulla[4]

1 Department of Psychology, Universidad Loyola Andalucía, Seville, Spain, 2 Department of Quantitative Methods, Universidad Loyola Andalucía, Seville, Spain, 3 Faculty of Brain Sciences, Division of Psychiatry, University College London, London, United Kingdom, 4 Centre for Mental Health Research, Research School of Population Health, ANU College of Health and Medicine, Australian National University, Canberra, Australia

* nmalmeda@uloyola.es

**Data Availability Statement:** The relevant data can be found here: https://datadryad.org/stash/share/

## Abstract

Rehabilitation services have a key role in ensuring integrated and comprehensive mental health (MH) care in the community for people suffering from long-term and severe mental disorders. MH-supported accommodation services aim to promote service users' autonomy and independence. Given the complexity associated with MH-supported accommodation services in England, a comparative evaluation of critical performance indicators, including service provision and quality of care, seems to be necessary in designing evidence-informed policies. This study aims to explore the influence of service quality indicators on the performance of MH-supported accommodation services in England. The analysed sample includes supported accommodation services from 14 nationally representative local authorities in England from the QuEST study grouped by three main types of care: residential care homes (divided into two subgroups: move-on and non-move-on oriented), supported housing and floating outreach. EDeS-MH (efficient decision support-mental health) was used to assess the performance indicators for the selected services by combining a Monte Carlo simulation engine, data envelopment analysis and a fuzzy inference engine for integrating expert knowledge. Depending on the type of care, six/seven quality domains were sequentially included after a baseline scenario (only technical) was analysed. Relative technical efficiency scores for the baseline scenarios revealed high performance in all the selected supported accommodation services, but the statistical variability was high. Quality domains significantly improved performance in every type of care. The inclusion of quality indicators has a positive impact on the global performance of each type of care. Remaining at the corresponding services more than expected for two years has a negative impact on performance. These findings can be considered from a planning perspective to facilitate the design of pathways of care with more realistic expectations about gaining autonomy in two years.

1eXNj4HhRp0ztCLfgzkwO53k7Rh8q0MvVeZh_A14-xs.

**Funding:** This study was funded by Junta de Andalucía Fondos FEDER (Unión Europea) (grant number: PY18-RE-0022; author who received the award: CGA) https://www.juntadeandalucia.es/economiaconocimientoempresasyuniversidad/fondoseuropeosenandalucia/feder. In addition, Instituto de Salud Carlos III funded this research study (grant number: PI18/01521; author who received the award: CGA) https://www.isciii.es/Paginas/Inicio.aspx. Finally, the Bupa Health Foundation – Australia also supported this study; author who received the award: LSC) https://www.bupa.com.au/about-us/bupa-health-foundation. There was no additional external funding received for this study. The funders had no role in study design, data collection and analysis, decision to publish, or preparation of the manuscript.

**Competing interests:** The authors have declared that no competing interests exist.

## Introduction

The Hospital Plan of 1962 in England established its position as a pioneer in the transition of psychiatric services from largely hospital-based settings to community-based mental health (MH) care [1]. Supported accommodation services are a key component of the 'theoretical whole system care pathway' that provides support in the community for people with more complex MH needs [2]. Ideally, these should be organized into a 'care pathway' with the expectation that people graduate from higher to lower levels of support over time. The QuEST (Quality and Effectiveness of Supported Tenancies for People with Mental Health Problems) study analysed supported accommodation services across England in 2014. This study identified three main types of services, residential care homes, supported housing and floating outreach, and investigated their quality and effectiveness. Residential care homes cater to those with the highest needs and include communal facilities staffed 24 hours that provide meals, medication supervision, cleaning, etc., where placements are not time-limited. Supported housing services provide shared or individual, self-contained, time-limited tenancies with staff based on-site up to 24 hours a day to help residents gain skills to move on to less supported accommodations. Finally, floating outreach services provide visiting support consisting of a few hours per week to people living in permanent, self-contained, individual tenancies with the aim of reducing support over time to zero. Users of residential care and supported housing are more likely to have a diagnosis of schizophrenia or other psychosis, whereas users of floating outreach are more likely to have a diagnosis of a common mental disorder, but the level of needs of people using supported housing and floating outreach is similar [3,4]. This study found that the quality of care was higher in MH-supported housing than in residential care or floating outreach services using multivariate statistical techniques. To increase its applicability and transferability, this analysis should be completed with standard information on service availability and modelling based on system dynamics and complexity [5].

The standard performance assessment of any kind of comparable service includes studying their input consumption and output production (directly related to the inputs consumed). These inputs/outputs are mainly technical, and in the end, the best service performance is always related to the most appropriate balance between the available inputs and the produced outputs. Researchers and decision-makers can seek to reduce (minimize) the amount of inputs for a given amount of outputs (input orientation) or vice versa to increase (maximize) the output production for a specific amount of inputs (output orientation).

The incorporation of quality variables in performance assessments (represented by six/seven quality domains in this study) is always complicated because these variables are subjective (perceived quality from users, managers, etc.). To our knowledge, this study is the first to include quality domains estimated by the managers of the selected MH services in a performance analysis [6]. Independent of the technical performance of the services (the baseline scenario analysed without quality domains), the main research objective is to assess the impact on service performance when quality domains are incorporated into the analysis. Considering that MH service managers have a specific amount of inputs and always try to obtain the corresponding best results, a neutral/positive relationship between managerial processes (input and output management) and their quality perceptions of performance is expected.

Relative technical efficiency (RTE) is a decision support measure that can be used to guide health informed evidence-based policy-making, mainly to improve resource allocation [7,8]. RTE assesses the relationship among the amount of inputs consumed and outputs produced by a set of comparable decision-making units [9]. RTE can be regarded as a synthetic meta-indicator that facilitates monitoring the evolution of a system and the dynamic relationships or connections across different performance indicators [10,11]. It has been used to identify

tailor-made improvement strategies for health care ecosystems such as the provision and resourcing of addiction treatment clinics [12,13], residential MH facilities [14], homes for people with mental disability [15], clinics for children and youth [16] and community-based youth services [17]. The RTEs of primary care and MH ecosystems have been systematically assessed in Basque County (Spain) [5,18–20]. Data envelopment analysis (DEA) has been widely used to assess the RTE of health services [6,18,21]. This group of nonparametric techniques is very robust and flexible because they do not need any preliminary assumption on the variable statistical structure, implying that variables (inputs/outputs) from different origins and types (e.g., the number of beds, technical input, the quality of care, manager perception) can be analysed at the same time. Quality variables have been included in DEA to assess the RTE of systems operating in different socioeconomic contexts (e.g., hospital care, schools or the banking industry) [22–25].

DEA can be included in a Monte Carlo simulation engine to include uncertainty and randomness in data values (all of which are considered statistical distributions) and design more realistic models [5]. As DEA is an operational model, it is completely blind. Variable values must be interpreted according to existing expert knowledge (usually a theoretical paradigm). A fuzzy inference engine allows us to operationalize the formalization of the balanced care model following the Expert-based Collaborative Analysis—EbCA methodology [5]. This engine interprets variable values before RTE was calculated.

This study aims to assess the impact of quality indicators (managerial perspective) on the performance (RTE) of selected MH-supported accommodation services in the English pathway of care. This objective includes the formalization of specific quality domains into variables (rates), their integration in RTE assessment, and a comparative impact analysis of quality variables on ecosystem performance to support decision-making and investment by providing relevant information for service managers to inform practice and service planning. Accordingly, this paper first presents a description of the ecosystem under study (a representative sample of supported accommodation services in England). Then, the selected variables are described and grouped into scenarios to highlight different perspectives of the ecosystem situation. Finally, the methodology used to assess ecosystem performance (including quality domains) is briefly described.

## Methods

### Setting

Data for supported accommodation services from 14 nationally representative local authorities in England were collected for the QuEST study [4]. Face-to-face interviews were conducted with service managers, key staff, and service users to assess the quality and characteristics of the services and those using them. The Quality Indicator for Rehabilitative Care–Supported Accommodation (QuIRC-SA) was completed with service managers. This standardized tool assesses service quality in seven domains: living environment, therapeutic environment, treatments and interventions, self-management and autonomy, social interface, human rights and recovery-based practice [26]. Data on the service's annual budget, weekly cost per user and service resources were also provided by the service managers to complement standard service costs for estimation of the cost-effectiveness of services [4]. The dataset is available at the Dryad digital repository (https://doi.org/10.5061/dryad.j0zpc86dz).

### Scenarios (variable grouping)

Measurement units, usually expressed as rates, for each variable were discussed by an expert group comprising senior clinicians, policy-makers, providers, and researchers with expertise

in MH-supported accommodation. Specifically, the experts who were involved included a rehabilitation psychiatrist who works with people living in MH-supported accommodations and is an international leader in this field, a psychiatrist with national-level expertise in policy pertaining to people with severe MH problems, two national leaders in MH-supported accommodation service provision and policy and three researchers who collected data from 87 supported accommodation services across England during the QuEST study. According to the background on evaluating the performance of MH-supported accommodation services [26–28], the relevant inputs were service budget (£ per place), places (number), and full-time equivalent staff (professionals per service user), and the relevant outputs were the average length of stay (years), occupied beds (%), the number of service users who moved to more independent accommodations (users per place), and the seven QuIRC-SA quality of care variables considering the service size (*value of the domain×available places*/100). By using the last mathematical transformation, original quality indicators considered the size of the service (an indicator value of 90 does not have the same meaning for a service with 10 places/beds and a service with 100 places/beds). All the considered transformations of original data render the selected services comparable by eliminating the potential "size" effect on performance assessments.

Eight different scenarios, or input/output variable combinations, were designed to assess the RTE of residential care and supported housing services (Table 1). For floating outreach services, only seven scenarios were identified because this type of care does not include the "living environment" QuIRC-SA service quality domain. Scenario 1 can be considered the "reference" scenario because it does not include any quality domains.

## Decision support system

An adaptation of the hybrid Decision Support System (DSS) EDeS-MH (Efficient Decision Support-Mental Health) was used [19] to assess the performance of the services. This computer-based tool included a Monte Carlo simulation engine, DEA and a fuzzy inference engine.

**Table 1. Descriptions of the scenarios for the RTE assessment of MH residential care, supported housing and floating outreach services.**

| MH supported accommodation services | Scenarios | Variables |
|---|---|---|
| Residential care and supported housing services | Scenario 1 (Baseline) | Inputs: N° of available beds or places, N° of available staff per service user, Annual budget per bed/place. Outputs: Years of stay in each service, Occupied beds/places (%), N° of service users who moved to a more independent accommodation per bed/place |
| | Scenario 2 | Baseline variables + QuIRC-SA living environment domain score |
| | Scenario 3 | Baseline variables + QuIRC-SA therapeutic environment domain score |
| | Scenario 4 | Baseline variables + QuIRC-SA self-management and autonomy domain score |
| | Scenario 5 | Baseline variables + QuIRC-SA social interface domain score |
| | Scenario 6 | Baseline variables + QuIRC-SA human rights domain score |
| | Scenario 7 | Baseline variables + QuIRC-SA treatments and interventions domain score |
| | Scenario 8 | Baseline variables + QuIRC-SA recovery-based practice domain score |
| Floating outreach services | Scenario 1 (Baseline) | Inputs: N° of available places, N° of available staff per service user, Annual budget per place. Outputs: Years of stay in each service, Occupied places (%), N° of service users who have moved from the service to another with greater independence per place. |
| | Scenario 2 | Baseline variables + therapeutic environment domain score |
| | Scenario 3 | Baseline variables + self-management and autonomy domain score |
| | Scenario 4 | Baseline variables + social interface domain score |
| | Scenario 5 | Baseline variables + human rights domain score |
| | Scenario 6 | Baseline variables + treatments and interventions domain score |
| | Scenario 7 | Baseline variables + recovery-based practice domain score |

The simulation engine was developed to address the uncertainty (data imprecision and vagueness) and randomness (unexpected facts) of real environments and to artificially multiply the number of observations [29]. The inner uncertainty of any ecosystem can be overcome by transforming original data values into statistical distributions (from standard datasets to statistical distribution bases). In each simulation, the Monte Carlo simulation engine analyses a new dataset selected at random. The statistical analysis of the final results (the process is stopped when the statistical error is lower than 2.5% for the mean) includes a sensitivity analysis of the ecosystem under study. The results (RTE scores) for each DMU and scenario are statistical distributions that can be studied in a more (basic statistics) or less (stability and entropy) standard manner [19,21]. The characteristics of these statistical distributions represent the potential reaction of the DMU to data changes.

DEA [9] was selected to evaluate the RTE of MH-supported accommodation services. This nonparametric technique has been widely used to assess health service performance [30] and, to a lesser extent, to evaluate MH services [6]. The standard DEA model is a linear programming one which structure is:

$$Min\theta - \varepsilon\left(\sum_{h=1}^{i} S_h^- + \sum_{r=1}^{j} S_r^+\right)$$

$$s.t.$$

$$\sum_{m=1}^{d} x_{hm}\lambda_m + S_h^- = \theta x_{ho}; h = 1, 2, ..., i$$

$$\sum_{m=1}^{d} y_{rm}\lambda_m - S_r^+ = y_{ro}; r = 1, 2, ..., j$$

$$\sum_{m=1}^{d} \lambda_m = 1$$

$$\lambda_m \geq 0; m = 1, 2, .., d$$

where $d$ is the number of DMU, $i$ the number of inputs and $j$ the number of outputs. The DMU $m$ consumes $x_{hm}$ of input $h$ and produces $y_{rm}$ of output $r$. $\theta$ is the efficiency score and $S_h^-$ and $S_r^+$ are the slacks [21].

In this research, the variable returns to scale DEA [31] was selected because when studying MH services, real output variations cannot be considered proportional to the corresponding input modifications [32], and constant returns to scale would involve a constant variation that cannot be considered realistic. The input-oriented DEA model was applied to assess whether service input consumption can be reduced while assuming a constant output level [9], which is especially relevant for decision-makers who must allocate finite resources to meet population needs. Output maximization (output-oriented DEA) is especially difficult and sometimes not recommendable (for example, when the system artificially tries to maximize the number of users who are moved to a service with greater independence, this can be mathematically correct but from a health care perspective it has no sense at all) when supported accommodation services are assessed. Finally, no weight control of the variables was considered. When the data envelopment analysis (DEA) did not include weight control of the inputs/outputs, a two-step process was followed to verify the impact of quality variable needs. First, RTE must be assessed without them, which yields 'operational results' (baseline). Second, quality variables must then be included as outputs to investigate the influence of quality on performance. Finally, when the number of observations is low (here, the number of move-on residential services is especially low), DEA cannot be sufficiently discriminative (in the end, the methodology tends to

show that all DMUs are efficient). However, as explained before, the uncertainty analysis (Monte Carlo simulation engine) multiplies the number of observations by the number of selected simulations, which overcomes this DEA drawback.

Prior to RTE assessment, variable (inputs and outputs) data values must be interpreted according to a preselected paradigm; otherwise, performance results can be biased. The fuzzy inference engine carries out this process automatically by including an expert-based rule base (IF . . . THEN . . .) according to the Balanced Care Model developed by Thornicroft and Tansella [33] and the pathway of care of MH-supported accommodation services [34]. This paradigm was used to define the range of adequacy for each variable. For example, the variable values for the service budget (£ per place)—input—in floating outreach services were considered adequate between the range [5000, 6000]. In this range, a greater value corresponds to better competence (it is mathematically transformed to be interpreted by the analytical procedure). Outside this range, the variable value is penalized by multiplying it by a parameter (in this specific case, by 2) because it is considered less adequate than when inside the range. The specific references for data value interpretation have been defined by a panel of experts according to the paradigm selected and their expertise. This process followed the EbCA model [5], where an iterative sequence of expert-based reviews culminates in a consensus. Once the references for interpreting variable values are defined, the EDeS-MH automatically runs a mathematical transformation based on an equation (linear monotone transformation) or a fuzzy operator (product-sum gravity method) to obtain the "transformed" value [21]. These transformed values will be analysed by DEA to determine the corresponding RTE scores (statistical distributions).

To evaluate the statistical significance of the differences between the baseline (without quality variables, scenario 1) and the rest of the designed scenarios (corresponding to each quality domain, scenarios 2 to 7/8), the nonparametric statistical Wilcoxon signed-rank test was used.

## Procedure

First, the types of services were coded according to the DESDE-LTC (Description and Evaluation of Services and DirectoriEs for Long Term Care) [35,36], an international service classification system that facilitates comparisons across different jurisdictions and studies and is used to describe service availability in our previous RTE studies using EDeS-MH. Second, expert-based cooperative analysis (EbCA) [5] was used to formalize expert knowledge on MH-supported accommodation services into a knowledge base structured by rules. This knowledge base facilitates interpretation of variable values according to the selected paradigm. EbCA has previously been used to evaluate the RTE of MH areas in Basque County [19].

The EbCA panel of experts on MH-supported accommodation services in England included persons in charge of the QuEST study, MH planners, MH managers and academic researchers (see the Scenario section for details). The interaction among these agents in the meetings was considered crucial for acquiring explicit their implicit knowledge: EbCA is an iterative procedure. The fuzzy inference engine interpreted variable values according to the resulting ranges of adequacy [19] according to the selected paradigm.

Services considered outliers in terms of the three main supported accommodation service groups were also identified by the EbCA panel. For example, some of the residential care services were oriented to work with people to helping them move on to more independent accommodations, whereas others worked with people who were likely to need high levels of support long term. Missing values (very few) were statistically imputed using a Monte Carlo simulation procedure based on the real statistical distribution of the corresponding variable.

Original data were randomized using symmetric triangular statistical distributions (5% variation on each side of the corresponding original value). This range includes feasible data variations (imprecision and vagueness). No critical stress on the ecosystem was included in the analysis (randomness). This procedure includes both data variations corresponding to ecosystem evolution (population, user mobility, etc.) and the effect of the time.

The Monte Carlo simulation engine selects a specific data value from the dataset in each computer run. Then, the fuzzy inference engine interprets these values according to the paradigm. The obtained RTE performance scores for each MH service are statistical distributions that can be analysed accordingly. RTE scores are always in a [0, 1] range. If RTE = 0, then the service is completely inefficient; if RTE = 1, then the service is efficient, and if the RTE score is between (0, 1), then the service is inefficient (a greater RTE score corresponds to higher service efficiency). For example, if a service has an RTE = 0.45 and another has an RTE = 0.98, both are inefficient, but the second service is very close to the efficiency.

## Results

### Basic statistics

Nine move-on residential services, 13 non-move-on residential services, 34 supported housing services and 30 floating outreach services were finally analysed because they have complete quality datasets. Five hundred simulations were run by the DSS. Basic statistics for the original data are shown in Table 2. The results show that the variability among services in each type of care is very high.

### Relative technical efficiency

The representativeness of the variables (inputs/outputs) for the RTE assessment was confirmed by the EbCA panel. Therefore, they can be considered indicators for real informed evidence-based decision-making. Quality domains (assessed by the QuIRC-SA [26]) were considered outcomes for performance assessments.

Only one service was considered an outlier and removed from the analysis. On the other hand, the analytical process of setting the measurement units for the variables highlighted the existence of two different groups in the residential care services dataset: move-on and non-move-on oriented services.

Regarding the move-on-oriented MH residential care services, the average RTE oscillated from 0.7, the worst, (scenario 1, baseline) to 0.86, the best, (scenario 2, therapeutic environment) (Table 3). In the baseline scenario, the highest performance (0.94) was achieved by service 22, while service 18 had the lowest score (0.5). Therefore, the performance of this type of care is very heterogeneous. When quality variables are included as outputs in scenarios 2–8, the RTE mean (which was always greater than 0.84) and the performance of each service increased significantly ($p < 0.001$) compared to the baseline. Services 4 and 26 had the worst performance.

In the non-move-on MH-oriented residential care services, the average RTE was 0.79, with service 90 (0.96) having the best and services 61 and 81 the worst RTE scores (Table 4). Again, including quality variables had a statistically significant ($p < 0.001$) positive impact (the performance increased), but the heterogeneity remained high. Scenarios 7 (treatments and interventions), 8 (recovery-based practices) and 4 (self-management and autonomy) were associated with the highest increase in performance. Services 61 and 81 showed low performance.

For MH-supported housing services, the average RTE oscillated from [0.54 to 0.64 (Table 5). In the baseline model, the average RTE was 0.54 (with service 86 (0.89) performing

**Table 2. Basic statistics for the variables used in the RTE assessment.**

| Type of supported accommodation | Basic statistics | Places | Total full time-equivalent professionals | Annual budget (£) | Length of stay (years) | Occupied places | Number of service users who have moved to a more independent accommodation over the last 2 years |
|---|---|---|---|---|---|---|---|
| Residential care (non-move on oriented) | Mean | 21.26 | 0.66 | 500,623.10 | 10 | 19.68 | 0.58 |
| | Standard deviation | 7.12 | 0.31 | 167,559.51 | 5.42 | 7.46 | 0.69 |
| | Variation coefficient (%) | 33.47 | 46.41 | 33.47 | 54.25 | 37.91 | 119.61 |
| | Minimum | 9 | 0.34 | 211,897.40 | 4 | 8 | 0 |
| | Maximum | 40 | 1.66 | 941,766.23 | 20 | 37 | 2 |
| Residential care (move on oriented) | Mean | 15.67 | 0.83 | 386,467.36 | 3.13 | 11.56 | 6.22 |
| | Standard deviation | 7.52 | 0.56 | 285,133.49 | 1.55 | 5.41 | 3.46 |
| | Variation coefficient (%) | 47.98 | 67.60 | 47.98 | 49.68 | 46.83 | 55.54 |
| | Minimum | 7 | 0.46 | 265,535.16 | 2 | 7 | 2 |
| | Maximum | 27 | 2.25 | 1,024,207.06 | 6 | 23 | 12 |
| Supported housing | Mean | 10.99 | 0.45 | 334,635.12 | 3.24 | 10.27 | 5.63 |
| | Standard deviation | 5.11 | 0.27 | 155,682.08 | 2.97 | 5.15 | 6.84 |
| | Variation coefficient (%) | 46.52 | 59.47 | 46.52 | 91.76 | 50.14 | 121.55 |
| | Minimum | 3 | 0.10 | 91,363.00 | 1 | 1 | 0 |
| | Maximum | 28 | 1.61 | 852,721.33 | 20 | 28 | 40 |
| Floating outreach | Mean | 29.97 | 0.17 | 171,950.08 | 2.83 | 28.89 | 13 |
| | Standard deviation | 22.90 | 0.17 | 131,354.68 | 2.16 | 23.02 | 16.49 |
| | Variation coefficient (%) | 76.39 | 103.51 | 76.39 | 76.45 | 79.69 | 126.85 |
| | Minimum | 5 | 0.03 | 28,685.67 | 1 | 4 | 0 |
| | Maximum | 80 | 0.97 | 458,970.67 | 9 | 6 | 75 |

the best and service 89 (0.33) performing the worst; the aggregation of quality variables increased the average RTE, and these differences were significant ($p < 0.001$). Here, scenario 7 (treatment and interventions) showed the lowest increase. Services 104, 4, 114, 70, 84, 19, 101,

**Table 3. Average relative technical efficiency scores for MH residential care services (move-on oriented).** Darker shading corresponds to lower RTE scores (less efficient scenarios and services).

| Services | Scenario 1 | Scenario 2 | Scenario 3 | Scenario 4 | Scenario 5 | Scenario 6 | Scenario 7 | Scenario 8 |
|---|---|---|---|---|---|---|---|---|
| 1 | 0.8087 | 0.8195 | 0.8036 | 0.8154 | 0.8293 | 0.8268 | 0.8181 | 0.8113 |
| 2 | 0.9050 | 0.9269 | 0.9388 | 0.9407 | 0.9460 | 0.9098 | 0.9393 | 0.9175 |
| 4 | 0.7608 | 0.7885 | 0.6651 | 0.6885 | 0.5773 | 0.7377 | 0.6084 | 0.6706 |
| 13 | 0.5199 | 0.9356 | 0.9519 | 0.9381 | 0.9446 | 0.9412 | 0.9377 | 0.9431 |
| 17 | 0.8052 | 0.9534 | 0.9520 | 0.9578 | 0.9615 | 0.9542 | 0.9660 | 0.9603 |
| 18 | 0.5039 | 0.8998 | 0.9342 | 0.9280 | 0.9327 | 0.9415 | 0.9246 | 0.9052 |
| 21 | 0.4834 | 0.9241 | 0.9271 | 0.9268 | 0.9246 | 0.9172 | 0.9188 | 0.9273 |
| 22 | 0.9402 | 0.9518 | 0.9467 | 0.9439 | 0.9484 | 0.9555 | 0.9511 | 0.9462 |
| 26 | 0.5372 | 0.5738 | 0.5559 | 0.5721 | 0.5106 | 0.5569 | 0.5709 | 0.5801 |
| **Global average** | **0.6960** | **0.8637** | **0.8528** | **0.8568** | **0.8417** | **0.8601** | **0.8483** | **0.8513** |

**Table 4. Average relative technical efficiency scores for MH residential care services (non-move-on oriented).** Darker shading corresponds to lower RTE scores (less efficient scenarios and services).

| Services | Scenario 1 | Scenario 2 | Scenario 3 | Scenario 4 | Scenario 5 | Scenario 6 | Scenario 7 | Scenario 8 |
|---|---|---|---|---|---|---|---|---|
| 96 | 0.6611 | 0.8188 | 0.8403 | 0.7780 | 0.8295 | 0.7195 | 0.7599 | 0.8764 |
| 108 | 0.9343 | 0.9377 | 0.9435 | 0.9421 | 0.9337 | 0.9335 | 0.9427 | 0.9434 |
| 90 | 0.6770 | 0.9637 | 0.9662 | 0.9688 | 0.9615 | 0.9627 | 0.9676 | 0.9665 |
| 61 | 0.3127 | 0.3361 | 0.3566 | 0.3925 | 0.2858 | 0.3952 | 0.3829 | 0.3804 |
| 124 | 0.6489 | 0.9194 | 0.8466 | 0.8942 | 0.8690 | 0.8832 | 0.8630 | 0.8874 |
| 91 | 0.6272 | 0.7421 | 0.7368 | 0.6887 | 0.7166 | 0.7345 | 0.8091 | 0.8006 |
| 14 | 0.6871 | 0.8825 | 0.8688 | 0.8693 | 0.9224 | 0.8660 | 0.8749 | 0.8558 |
| 12 | 0.8610 | 0.8400 | 0.8734 | 0.8831 | 0.8248 | 0.8625 | 0.9091 | 0.8815 |
| 81 | 0.3878 | 0.4752 | 0.4269 | 0.5140 | 0.3965 | 0.4560 | 0.4863 | 0.4652 |
| 79 | 0.7367 | 0.8377 | 0.8688 | 0.9107 | 0.8321 | 0.7940 | 0.9261 | 0.8709 |
| 80 | 0.8279 | 0.8574 | 0.8747 | 0.8897 | 0.8101 | 0.8540 | 0.9157 | 0.8788 |
| 88 | 0.8342 | 0.9197 | 0.9167 | 0.9274 | 0.8884 | 0.8901 | 0.9026 | 0.9378 |
| 83 | 0.7272 | 0.7334 | 0.7966 | 0.8355 | 0.7680 | 0.8161 | 0.8240 | 0.7830 |
| **Global average** | **0.6659** | **0.7895** | **0.7935** | **0.8072** | **0.7722** | **0.7821** | **0.8126** | **0.8098** |

113, 60, 52, 89 and 95 had an average RTE lower than 0.5. The performance variability was very high in all scenarios.

In MH floating outreach services, the average RTE was in the 0.5 to 0.65 range (Table 6). For the other two types of supported accommodations, the baseline model showed the worst performance, and the inclusion of quality variables significantly increased both global and individual average service RTE ($p < 0.001$). Services 1, 63, 37 and 35 had very high performance. The heterogeneity of the sample was also very high.

Generally, the inclusion of quality domains highlights a neutral-positive or positive impact on the performance of supported accommodation services in England, which is especially relevant in some services that showed a lower average RTE: 13, 18 and 21 (33.3% of the surveyed services, Table 3); 14, 90, 96 and 124 (30.8% of the surveyed services, Table 4); 18, 22, 27, 58, 70, 97, 110 and 115 (23.5% of the surveyed services, Table 5); and, finally, 1, 7, 36, 56, 87, 109, 121 and 122 (26.7% of the surveyed services, Table 6). In these services, the manager´s perception of quality provided by their services surpasses technical results (baseline scenario), probably because of non-evaluated variables or circumstances in care provision.

On the other hand, quality variables decrease service performance only in supported accommodation services 15, 31, 32, 86 and 98 (14.7% of the surveyed services, Table 5). This behaviour can be considered strange because here, the manager´s perception of quality provided by their services underestimates their own technical results. Again, additional non assessed variables or circumstances can affect how the manager processes information with respect to detecting frameworks of improving quality of care.

Services where the inclusion of quality variables results in a neutral behaviour of their performance constituted the majority, respectively: 66.7%, 69.2%, 61.8% and 73.3% of the surveyed services. In these cases, the manager´s perception of quality provided by their corresponding services matches their technical results; they obtain a quality according to their resources and outcomes.

## Discussion

This study aimed to assess the impact of quality domains (formalized as variables) on the performance (RTE) of selected MH-supported accommodation services in the English pathway of

**Table 5. Average relative technical efficiency scores for MH-supported housing services.** Darker shading means lower RTE scores (less efficient scenarios and services).

| Services | Scenario 1 | Scenario 2 | Scenario 3 | Scenario 4 | Scenario 5 | Scenario 6 | Scenario 7 | Scenario 8 |
|---|---|---|---|---|---|---|---|---|
| 86 | 0.8926 | 0.3546 | 0.3274 | 0.3383 | 0.3375 | 0.3401 | 0.3235 | 0.3344 |
| 82 | 0.6868 | 0.7440 | 0.7718 | 0.7749 | 0.7488 | 0.7490 | 0.6777 | 0.7860 |
| 31 | 0.6297 | 0.4945 | 0.5212 | 0.4823 | 0.4942 | 0.4835 | 0.4754 | 0.5159 |
| 104 | 0.4839 | 0.4183 | 0.4017 | 0.4190 | 0.4309 | 0.4500 | 0.3947 | 0.3995 |
| 112 | 0.5472 | 0.5537 | 0.5802 | 0.6076 | 0.6042 | 0.5489 | 0.5890 | 0.5804 |
| 4 | 0.3665 | 0.3419 | 0.3357 | 0.3519 | 0.3454 | 0.3478 | 0.3351 | 0.3453 |
| 116 | 0.5683 | 0.5304 | 0.5963 | 0.6253 | 0.5633 | 0.6085 | 0.4972 | 0.6324 |
| 111 | 0.5646 | 0.5977 | 0.6530 | 0.6680 | 0.7163 | 0.6524 | 0.5991 | 0.7021 |
| 123 | 0.4765 | 0.5371 | 0.5397 | 0.5406 | 0.5411 | 0.5352 | 0.5390 | 0.5365 |
| 114 | 0.3522 | 0.3338 | 0.3410 | 0.3496 | 0.3274 | 0.3330 | 0.3197 | 0.3264 |
| 22 | 0.6278 | 0.8948 | 0.7238 | 0.7906 | 0.9223 | 0.7703 | 0.6706 | 0.7024 |
| 58 | 0.5488 | 0.8771 | 0.8960 | 0.8708 | 0.9384 | 0.9351 | 0.8678 | 0.8854 |
| 18 | 0.5647 | 0.7974 | 0.8075 | 0.8235 | 0.8210 | 0.8570 | 0.8281 | 0.8437 |
| 70 | 0.3385 | 0.8997 | 0.8961 | 0.9064 | 0.8955 | 0.9008 | 0.9057 | 0.9011 |
| 84 | 0.5090 | 0.5922 | 0.5926 | 0.5994 | 0.5896 | 0.6003 | 0.5971 | 0.5930 |
| 103 | 0.3390 | 0.3570 | 0.3463 | 0.3432 | 0.3417 | 0.3386 | 0.3415 | 0.3543 |
| 15 | 0.7464 | 0.5751 | 0.5692 | 0.5784 | 0.5747 | 0.5754 | 0.5710 | 0.5765 |
| 19 | 0.3554 | 0.3591 | 0.3608 | 0.3605 | 0.3416 | 0.3457 | 0.3476 | 0.3568 |
| 101 | 0.3737 | 0.3815 | 0.4270 | 0.3735 | 0.4388 | 0.4223 | 0.3818 | 0.4112 |
| 113 | 0.3551 | 0.5565 | 0.6863 | 0.6480 | 0.7425 | 0.5846 | 0.6218 | 0.6705 |
| 110 | 0.5134 | 0.7842 | 0.9469 | 0.9396 | 0.9095 | 0.9333 | 0.9210 | 0.9491 |
| 115 | 0.5765 | 0.8830 | 0.9059 | 0.8638 | 0.9005 | 0.8686 | 0.8912 | 0.8708 |
| 98 | 0.7271 | 0.5683 | 0.5623 | 0.5593 | 0.5608 | 0.5742 | 0.5602 | 0.5630 |
| 60 | 0.3543 | 0.3404 | 0.3418 | 0.3293 | 0.3609 | 0.3507 | 0.3239 | 0.3380 |
| 52 | 0.3620 | 0.3613 | 0.3401 | 0.3405 | 0.3277 | 0.3329 | 0.3393 | 0.3304 |
| 41 | 0.9523 | 0.9323 | 0.9288 | 0.9333 | 0.9306 | 0.9342 | 0.9274 | 0.9309 |
| 105 | 0.5374 | 0.6729 | 0.7444 | 0.7426 | 0.8129 | 0.7595 | 0.7414 | 0.7572 |
| 27 | 0.5059 | 0.9600 | 0.9578 | 0.9605 | 0.9438 | 0.9536 | 0.9569 | 0.9580 |
| 46 | 0.6588 | 0.7817 | 0.8252 | 0.8612 | 0.9432 | 0.8276 | 0.6807 | 0.9087 |
| 93 | 0.9060 | 0.9194 | 0.9229 | 0.9191 | 0.9137 | 0.9186 | 0.9087 | 0.9165 |
| 89 | 0.3301 | 0.3428 | 0.3447 | 0.3450 | 0.3528 | 0.3407 | 0.3407 | 0.3189 |
| 95 | 0.3389 | 0.5607 | 0.5321 | 0.5548 | 0.5857 | 0.5249 | 0.5241 | 0.5724 |
| 32 | 0.6847 | 0.3679 | 0.3691 | 0.3706 | 0.3770 | 0.3696 | 0.3509 | 0.3719 |
| 97 | 0.4565 | 0.9577 | 0.9565 | 0.9326 | 0.9614 | 0.9633 | 0.9629 | 0.9424 |
| **Global average** | **0.5362** | **0.6067** | **0.6192** | **0.6207** | **0.6352** | **0.6185** | **0.5974** | **0.6230** |

care. To our knowledge, this study is the first to evaluate the performance of MH-supported accommodation services with the inclusion of service quality indicators in the analyses. The integration of the quality construct (structured by a set of specific domains with varying relevance) in DEA is complex. Authors have addressed this issue using different techniques: quality-adjusted DEA (Q-DEA) [37], the two-model approach [38], quality and operating efficiency models with weight restrictions [39], and the multiple objective approach to DEA (MODEA) [40]. The EDeS-MH decision support system has been successfully adapted to assess MH services instead of catchment areas [5,18,19,21]. This DSS transfers the robust methods for resource allocation and health technology assessment used at national (macro) and regional (macro) levels to the local (meso) and service (micro) levels, reducing the lack of

**Table 6. Average relative technical efficiency scores for MH floating outreach services.** Darker shading corresponds to lower RTE scores (less efficient scenarios and services).

| Services | Scenario 1 | Scenario 2 | Scenario 3 | Scenario 4 | Scenario 5 | Scenario 6 | Scenario 7 |
|---|---|---|---|---|---|---|---|
| 92 | 0.5831 | 0.6022 | 0.6095 | 0.5960 | 0.6024 | 0.5912 | 0.6134 |
| 85 | 0.3387 | 0.3429 | 0.3562 | 0.3362 | 0.3235 | 0.3383 | 0.3371 |
| 102 | 0.4471 | 0.4961 | 0.4953 | 0.4641 | 0.4791 | 0.4808 | 0.4877 |
| 87 | 0.5476 | 0.7281 | 0.7072 | 0.6431 | 0.7068 | 0.7233 | 0.7127 |
| 107 | 0.5184 | 0.6184 | 0.6251 | 0.6001 | 0.5956 | 0.6455 | 0.6230 |
| 121 | 0.4610 | 0.7668 | 0.8374 | 0.9373 | 0.7397 | 0.7918 | 0.8087 |
| 122 | 0.5104 | 0.9619 | 0.9610 | 0.9553 | 0.9644 | 0.9581 | 0.9649 |
| 109 | 0.4821 | 0.9539 | 0.9627 | 0.9640 | 0.9542 | 0.8079 | 0.9651 |
| 118 | 0.3645 | 0.4149 | 0.4136 | 0.3756 | 0.3895 | 0.4193 | 0.4154 |
| 119 | 0.3498 | 0.3558 | 0.3532 | 0.3343 | 0.3443 | 0.3447 | 0.3471 |
| 1 | 0.7258 | 0.9724 | 0.9649 | 0.9441 | 0.9603 | 0.9705 | 0.9695 |
| 106 | 0.3375 | 0.3542 | 0.3460 | 0.3438 | 0.3513 | 0.3430 | 0.3397 |
| 71 | 0.3544 | 0.4078 | 0.4136 | 0.4107 | 0.4067 | 0.4200 | 0.4147 |
| 16 | 0.5384 | 0.6061 | 0.6091 | 0.5883 | 0.6038 | 0.6071 | 0.6060 |
| 63 | 0.9518 | 0.9484 | 0.9492 | 0.9468 | 0.9587 | 0.9482 | 0.9485 |
| 94 | 0.4697 | 0.7562 | 0.7906 | 0.6975 | 0.6792 | 0.8015 | 0.7964 |
| 65 | 0.4517 | 0.4820 | 0.4971 | 0.5094 | 0.4835 | 0.4755 | 0.4798 |
| 7 | 0.4958 | 0.9096 | 0.8757 | 0.9434 | 0.8519 | 0.9649 | 0.8707 |
| 8 | 0.4834 | 0.5018 | 0.5109 | 0.5067 | 0.5024 | 0.5012 | 0.5187 |
| 40 | 0.5113 | 0.7406 | 0.7538 | 0.6752 | 0.7327 | 0.7049 | 0.7839 |
| 99 | 0.3348 | 0.3689 | 0.3800 | 0.3596 | 0.3460 | 0.3556 | 0.3715 |
| 100 | 0.4855 | 0.6292 | 0.6457 | 0.6697 | 0.6147 | 0.6231 | 0.6631 |
| 56 | 0.4920 | 0.7355 | 0.7241 | 0.6048 | 0.6881 | 0.7131 | 0.7712 |
| 51 | 0.3438 | 0.5738 | 0.4857 | 0.4161 | 0.6451 | 0.7797 | 0.7535 |
| 76 | 0.3658 | 0.3620 | 0.3736 | 0.3545 | 0.3361 | 0.3606 | 0.3597 |
| 78 | 0.4137 | 0.4001 | 0.4149 | 0.4035 | 0.4138 | 0.4219 | 0.4009 |
| 77 | 0.3523 | 0.3532 | 0.3533 | 0.3567 | 0.3404 | 0.3501 | 0.3459 |
| 36 | 0.3443 | 0.8132 | 0.8580 | 0.8939 | 0.8755 | 0.4439 | 0.8841 |
| 37 | 0.9317 | 0.9366 | 0.9339 | 0.9370 | 0.9304 | 0.9396 | 0.9427 |
| 35 | 0.8628 | 0.8599 | 0.8523 | 0.8563 | 0.8596 | 0.8654 | 0.8675 |
| **Global average** | **0.4950** | **0.6318** | **0.6351** | **0.6208** | **0.6227** | **0.6230** | **0.6454** |

transparency and accountability in relation to resource management, which often occurs at this level [41]. Finally, the EbCA highlights the existence of two different residential services: move-on and non-move-on oriented services, which must be studied separately.

To date, MH ecosystem performance has been assessed by combining operational and technical variables without including quality indicators [6]. Quality of care scores were determined by a standardized instrument applied through interviews with service managers; therefore, they are managerial perceptions. However, the ratings produced have been shown to correlate well with service user experiences of care [42].

Expert knowledge, formalized in a knowledge base, was crucial for interpreting the level of adequacy of variable values. For this process, selecting the appropriate paradigm was mandatory to overcome the limitations of classical DEA, where less input (resources) consumption, given an output level, was related to higher efficiency. In this research, the selected paradigm was structured by the Balanced Care Model [33] and the English pathway of care in MH-supported accommodation services [34]. The knowledge base is the core of the fuzzy inference

engine because it provides expert-based information for transforming the original data values (from the Monte Carlo simulation engine) according to the balanced care model. This transformation is based on experts' opinions on the adequacy of each data value and represents a specific framework that must change throughout the time span and depend on the socioeconomic environment. Therefore, the same dataset (raw data) can result in different results according to experts' perceptions of reality. The fuzzy inference engine can "understand" the nuances that decision-makers assume in system management.

A recent national survey of MH-supported accommodation services across England found that the quality of care was higher in MH-supported housing than in residential care or floating outreach services [4]. Considering that quality variables have not been included in any RTE assessment of MH services and ecosystems [6,13,15,18,43], the question to answer here is whether the perception (given by the corresponding managers) about the quality provided by an MH service is consistent with its technical performance. If the impact on RTE scores is negative, then the perception of the quality provided is not aligned with the technical performance (baseline RTE). In this situation, technical results do not achieve an equivalent quality according to the manager´s opinion. If the impact can be considered neutral, then technical results and quality perception are balanced. Finally, if it is positive, managers know that the quality provided by their services is better than that of the corresponding neutral impact and probably manage other variables that could not be gathered in the QuEST study.

The incorporation of quality domains as variables (outputs) in DEA had a neutral-positive or positive global impact on the performance of MH-supported accommodation services. The MH residential care services dataset was divided into those aiming to promote autonomy among service users (move-on oriented) and those focused on providing residential care (non-move-on oriented). This fact was highlighted by the EbCA panel of experts and allowed DEA to avoid the bias mainly induced by the interpretation of a critical output: "the number of service users who have moved to more independent accommodation". Instead of being the most expensive and probably complex services, RTE scores revealed relatively high performance in both groups (especially when quality variables were included), but the variability was high, revealing probable differences in management strategies.

Quality variables increased (sometimes not significantly) the average RTE in a relevant number of the selected services, but most showed a neutral-positive profile. Quality indicators included nontechnical but real service characteristics estimated by the respective managers that ultimately improved service performance scores.

The average RTE of MH-supported housing services was lower than that of residential care homes, and the input/output balance was slightly worse because of the higher statistical variability. This variability can be associated with differences in service users' needs and personal characteristics. The incorporation of quality variables significantly increased service performance and confirms the conclusions of a previous study by Killaspy et al. [4]: supported housing is a cost-effective type of care, as it provides support while promoting autonomy. Nevertheless, service performance scores were lower than expected due to the high variability among services and the low number of service users who moved to more independent accommodations (a critical technical output for the paradigm). A considerable number of services in MH-supported housing had an average RTE below 0.5 (relatively low performance), indicating that these types of services adjust their structures and obviously obtain different outcomes considering users' needs (tailor-made structures). For these specific services, quality variables did not significantly increase performance. Again, considering that quality was assessed by the service managers (managerial perspective), they implicitly stated that in these services, major changes can be developed. The English pathway of care is not a one-way road, and service users can be moved from one supported housing service to another (instead of to a floating

outreach service) or even remain in the same place longer than expected. This variable (N° of service users who moved to a more independent accommodation per bed/place) is likely to be outside the service's control to some degree, i.e., the service may be working hard with service users to help them gain skills to move on, but the supply of more independent accommodations where they can move on may be insufficient. In this type of care, a relatively pertinent number of services decreases performance scores when quality variables are included, which may indicate that managers consider other variables or service characteristics that lead to poorer quality results than they were expecting considering the corresponding resource level and outcome production.

The performance of MH floating outreach services was the lowest on average, but the incorporation of quality variables still led to a statistically significant increase in the average global RTE. The results reported by Killaspy et al. [4] also showed that these services provided lower-quality care than supported housing or residential care, but they also achieved the highest rate of movement to more independent accommodations. The high statistical variability of the services in this type of care indicates that they may be struggling to meet a wide range of service user needs. After adjusting for differences in patient characteristics, the move-on characteristic was a critical variable considering the Balanced Care Model (a paradigm for interpreting variable values). As occurs in residential houses, a relevant number of floating outreach services significantly increased their performance scores. Again, this increase is not aligned with the service management (resources and outcomes) shown in the baseline scenario.

MH service names were not revealed in this study to maintain data privacy.

## Conclusions

The ideal integration of supported accommodation services in an MH care pathway is unfortunately difficult to determine outside of highly integrated care systems. The existence of functioning care pathways is a major example of integration. With increasing integration, the inclusion of service quality variables in the RTE assessment of MH-supported accommodation services in England has become possible. Quality increased global service performance in the three types of care: residential care, supported housing and floating outreach. This neutral-positive or positive impact showed that RTE assessment using only technical variables is not sufficient to achieve a holistic view of service performance.

Expert knowledge formalization was critical for distinguishing the types of service, identifying outliers and interpreting variable values according to a specific care paradigm. EbCA demonstrated its practical utility when appropriate experts were available to join the panel.

The adaptation of the EDeS-MH to include quality variables required a two-step process. The results allowed us to have a better understanding of the performance of individual services and the supported accommodation care pathway. This approach may have utility in designing tailor-made improvement strategies for specific services as well as in service planning. The input/output balance of MH residential care services was appropriate because they were more structured from a managerial point of view (user needs are usually very well defined), while supported housing and floating outreach could be improved. These types of care are not structured due to the diversity of user needs associated with their higher level of autonomy compared to residential care service users. Considering that the English pathway of MH-supported accommodation is not a one-way road, the impact of remaining in supported housing and floating outreach services longer than the usual two years appears to have a major negative influence on service performance.

Services showing a significant increase or decrease in their performance scores when quality variables are included in the analysis should be studied. The observed differences from the

baseline scenario (only technical) must be a consequence of something related to specific structural or managerial characteristics. These characteristics are a crucial source of information for the design of new interventions, policies or strategies to improve MH care.

Future research including other components of the whole system (such as inpatient, day and outpatient care) is recommended to understand global MH ecosystem performance in England. By selecting service benchmarks, key variables requiring improvement can be identified to design specific policies and interventions. The integration of the user's satisfaction construct into the analysis is also a major trend.

## Acknowledgments

We would like to thank the experts of the EbCA nominal group: Jed Boardman, Andrew van Doorn, Kathy Roberts, Peter McPherson, Christian Dalton-Locke and Sarah Dowling.

## Author Contributions

**Conceptualization:** Nerea Almeda, Carlos Ramón García-Alonso.

**Data curation:** Nerea Almeda, Carlos Ramón García-Alonso, Helen Killaspy.

**Formal analysis:** Nerea Almeda, Carlos Ramón García-Alonso.

**Funding acquisition:** Carlos Ramón García-Alonso.

**Investigation:** Nerea Almeda, Luis Salvador-Carulla.

**Methodology:** Carlos Ramón García-Alonso.

**Resources:** Nerea Almeda.

**Software:** Carlos Ramón García-Alonso.

**Supervision:** Nerea Almeda, Mencía R. Gutiérrez-Colosía.

**Validation:** Nerea Almeda, Helen Killaspy.

**Writing – original draft:** Nerea Almeda, Mencía R. Gutiérrez-Colosía.

**Writing – review & editing:** Nerea Almeda, Carlos Ramón García-Alonso, Helen Killaspy, Mencía R. Gutiérrez-Colosía, Luis Salvador-Carulla.

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
