## [Editor Report · Decision Letter 0]

14 Jun 2021

PONE-D-21-17723

The critical factor: the role of quality in the performance of supported accommodation services for complex mental illness in England.

PLOS ONE

Dear Dr. Almeda,

Thank you for submitting your manuscript to PLOS ONE. After careful consideration, we feel that it has merit but does not fully meet PLOS ONE’s publication criteria as it currently stands. Therefore, we invite you to submit a revised version of the manuscript that addresses the points raised during the review process.

We look forward to receiving your revised manuscript.

Kind regards,

Dragan Pamucar

Academic Editor

PLOS ONE

Journal Requirements:

Additional Editor Comments:

Abstract

The abstract is loosely written. A standard abstract must present, without leaving any doubt, the objective of the paper precisely; source of data (which is not present in your abstract) and analytical approach used; key findings and any policy implication and recommendations.

Introduction

• The arguments are fairly presented but the statement that justifies the study does not come clearly (i.e. Why did you started this research?).

• The introduction does not precisely construct the research problem tackled and does not show how the problem is taken care.

• Why did you used MC simulation, DEA and fuzzy system in the study? These techniques also should be discussed.

• The research hypotheses’ are not mentioned in the introduction or clear in the literature review.

Literature review and critical analysis of theories, practices or commentary focusing on existing

documents

• The study lacks clear description of the literature review:

• What I am missing is a description of the review. Did you conduct a systematic literature review? Which years? Key words? What was the literature you found?

• Can you better describe how you came to your major variables? You have them from the literature review, but how was literature screened to derive these factors.

Methodology and scope of work

The analytical design is well prepared.

Results

The author has poorly discussed the results of the paper. One would expect to find the previous empirical work enriching the discussions of the results, but unfortunately, that has not been done.
---

## [Author Response · Author response to Decision Letter 0]

29 Jul 2021

Dear Editor,

Many thanks for your comments, we highly appreciate your suggestions and questions. According to them, we have carried out the corresponding changes in the paper (highlighted in yellow).

Sincerely,

The authors.

EDITOR’S COMMENTS

Comment nº 1.

The abstract is loosely written. A standard abstract must present, without leaving any doubt, the objective of the paper precisely; source of data (which is not present in your abstract) and analytical approach used; key findings and any policy implication and recommendations.

Thank you for your recommendation. A new abstract has been included following the proposed structure.

Comment nº 2.

The arguments are fairly presented but the statement that justifies the study does not come clearly (i.e. Why did you started this research?).

According to your kind suggestion, the following paragraph has been included in the introduction:

The standard performance assessment of any kind of comparable services includes input consumption and output production (directly related to the inputs consumed). These inputs/outputs are mainly technical. The incorporation of quality variables (in this study representing by six/seven quality domains) is always complicated because they are perceptions (perceived quality from users, managers, etc.). To our knowledge, this is the first study that includes quality domains, estimated by the managers of the selected MH services, into a performance analysis (6). Independently of the technical performance of the services (the baseline scenario analysed without quality domains) the main research question is to assess the impact on the service performance when quality domains are incorporated into the analysis. Taking into account that MH service managers have a specific amount of inputs and always try to obtain the corresponding best results, a neutral/positive relationship between managerial processes (input and output management) and their quality perceptions on performance is expected.

Other changes include a relevant modification of the following paragraph:

Relative Technical Efficiency (RTE) is a decision support technique that can be used for guiding health informed evidence-based policy-making, mainly on improving resources allocation (7,8). RTE assesses the relationship among the amount of inputs consumed and outputs produced by a set of comparable decision making units (9). RTE can be regarded as a synthetic meta-indicator that facilitates monitoring the evolution of a system and the dynamic relationships or connections across different performance indicators (10,11). It has been used for identifying tailor-made improvement strategies for health care ecosystems like provision and resourcing of addiction treatment clinics (12,13), residential MH facilities (14), homes for people with mental disability (15), clinics for children and youth (16) and community based youth services (17). The RTE of primary care and MH ecosystems have been systematically assessed in the Basque Country (Spain) (5,18–20). Data Envelopment Analysis (DEA) has been widely used to assess the RTE of health services (6,18,21). This group of non-parametric techniques are very robust and flexible because they don´t need any preliminary assumption on the variable statistical structure. This means that variables (inputs/outputs) from different origins and types (e.g. number of beds, technical input, and quality of care, manager perception) can be analysed at the same time. Quality variables have been included in DEA for assessing the RTE of systems operating in different socio-economic contexts (e.g. hospital care, schools or the banking industry) (22–25). 

DEA can be included in a Monte Carlo simulation engine to include uncertainty and randomness in data values (all of them are considered statistical distributions) and design more realistic models (5). Being DEA an operational model, it is completely blind. Variable values must be interpreted according to the existing expert knowledge (usually a theoretical paradigm). A fuzzy inference engine allows to make operational the formalization of the Balanced Care model following the EbCA methodology (5). This engine interprets variable values before RTE was calculated.

Comment nº 3.

The introduction does not precisely construct the research problem tackled and does not show how the problem is taken care.

Thank you for your comment. For that reason, we have included the following paragraph (see also the previous comment):

The standard performance assessment of any kind of comparable services includes input consumption and output production (directly related to the inputs consumed). These inputs/outputs are mainly technical. The incorporation of quality variables (in this study representing by six/seven quality domains) is always complicated because they are perceptions (perceived quality from users, managers, etc.). To our knowledge, this is the first study that includes quality domains, estimated by the managers of the selected MH services, into a performance analysis (6). Independently of the technical performance of the services (the baseline scenario analysed without quality domains) the main research question is to assess the impact on the service performance when quality domains are incorporated into the analysis. Taking into account that MH service managers have a specific amount of inputs and always try to obtain the corresponding best results, a neutral/positive relationship between managerial processes (input and output management) and their quality perceptions on performance is expected.

Comment nº 4.

Why did you used MC simulation, DEA and fuzzy system in the study? These techniques also should be discussed.

Thanks again, according to this comment we have modify the following paragraph in the introduction (see also comment 2):

Relative Technical Efficiency (RTE) is a decision support technique that can be used for guiding health informed evidence-based policy-making, mainly on improving resources allocation (7,8). RTE assesses the relationship among the amount of inputs consumed and outputs produced by a set of comparable decision making units (9). RTE can be regarded as a synthetic meta-indicator that facilitates monitoring the evolution of a system and the dynamic relationships or connections across different performance indicators (10,11). It has been used for identifying tailor-made improvement strategies for health care ecosystems like provision and resourcing of addiction treatment clinics (12,13), residential MH facilities (14), homes for people with mental disability (15), clinics for children and youth (16) and community based youth services (17). The RTE of primary care and MH ecosystems have been systematically assessed in the Basque Country (Spain) (5,18–20). Data Envelopment Analysis (DEA) has been widely used to assess the RTE of health services (6,18,21). This group of non-parametric techniques are very robust and flexible because they don´t need any preliminary assumption on the variable statistical structure. This means that variables (inputs/outputs) from different origins and types (e.g. number of beds, technical input, and quality of care, manager perception) can be analysed at the same time. Quality variables have been included in DEA for assessing the RTE of systems operating in different socio-economic contexts (e.g. hospital care, schools or the banking industry) (22–25). 

Additionally, we have modified the following paragraph:

DEA can be included in a Monte Carlo simulation engine to include uncertainty and randomness in data values (all of them are considered statistical distributions) and design more realistic models (5). Being DEA an operational model, it is completely blind. Variable values must be interpreted according to the existing expert knowledge (usually a theoretical paradigm). A fuzzy inference engine allows to make operational the formalization of the Balanced Care model following the EbCA methodology (5). This engine interprets variable values before RTE was calculated.

Comment nº 5.

The research hypotheses’ are not mentioned in the introduction or clear in the literature review.

Based on your comment, here you can see an additional explanation:

Based on original data from the QUEST project (Killaspy et al., 2017) and a previous systematic review about DEA models in mental health (Garcia-Alonso et al., 2019), this study tries to check, if it exists, the relationship between “… managerial processes (input and output management) and their -mental health service managers- quality perceptions on performance …”.

Taking into account this comment and according to previous ones, we have included the following paragraph:

The standard performance assessment of any kind of comparable services includes input consumption and output production (directly related to the inputs consumed). These inputs/outputs are mainly technical. The incorporation of quality variables (in this study representing by six/seven quality domains) is always complicated because they are perceptions (perceived quality from users, managers, etc.). To our knowledge, this is the first study that includes quality domains, estimated by the managers of the selected MH services, into a performance analysis (6). Independently of the technical performance of the services (the baseline scenario analysed without quality domains) the main research question is to assess the impact on the service performance when quality domains are incorporated into the analysis. Taking into account that MH service managers have a specific amount of inputs and always try to obtain the corresponding best results, a neutral/positive relationship between managerial processes (input and output management) and their quality perceptions on performance is expected.

Comment nº 6.

Literature review and critical analysis of theories, practices or commentary focusing on existing documents. The study lacks clear description of the literature review:

Thanks again, we have modified the introduction trying to make it clearer:

1. Explaining the relevance of the situation.

2. Introducing the relevance of integrating quality variables into the technical performance analysis of mental health services and introducing our main hypothesis.

3. Exploring the techniques in an integrated way.

4. Introducing the objective in a clearer way, as follows:

This study aims to assess the impact of quality indicators (managerial perspective) in the performance (RTE) of selected MH supported accommodation services in the English pathway of care. This objective includes the formalization of specific quality domains into variables (rates), their integration in RTE assessment, and a comparative impact analysis of quality variables on the ecosystem performance to support decision-making and investment by providing relevant information for service managers to inform practice and service planning.

In all of these sections, relevant literature has been reviewed and explained.

Comment nº 7.

What I am missing is a description of the review. Did you conduct a systematic literature review? Which years? Key words? What was the literature you found?

Sorry again, we have modified the objective of the paper because the introduction offered a wrong perspective of our analysis:

This study aims to assess the impact of quality indicators (managerial perspective) in the performance (RTE) of selected MH supported accommodation services in the English pathway of care. This objective includes the formalization of specific quality domains into variables (rates), their integration in RTE assessment, and a comparative impact analysis of quality variables on the ecosystem performance to support decision-making and investment by providing relevant information for service managers to inform practice and service planning.

No systematic review was conducted for the present paper.

Comment nº 8.

Can you better describe how you came to your major variables? You have them from the literature review, but how was literature screened to derive these factors. Methodology and scope of work.

The original dataset was collected from the QuEST study (raw data). In order to compare the selected MH services original data have been transformed into rates in order to eliminate the “size” effect. For example, it is not the same to have a budget of 100,000 pounds if you have 10 or 100 places or beds. All these rates were discussed several times by experts according to the EbCA model. 

The following sentence has been included in the “Scenarios” section:

All the considered transformations of original data make the selected services comparable by eliminating the potential “size” effect on performance assessment.

The rates included in this study provide a high reliability of measurement. Additionally, the large number of previous attempts to reach optimal representation of the data allowed us to find differences between residential services: move on and non-move on oriented. According to this, we have included the following sentence in the “Relative Technical Efficiency” section.

On the other hand, the analytical process of setting the measurement units for the variables highlighted the existence of two different groups in the residential care services dataset: move on and non-move on oriented.

Comment nº 9.

The author has poorly discussed the results of the paper. One would expect to find the previous empirical work enriching the discussions of the results, but unfortunately, that has not been done.

According to your kind suggestion we have added additional comments in the “Results” section:

Globally speaking, the inclusion of quality domains highlights a neutral-positive or positive impact on the performance of supported accommodation services in England. This is especially relevant in some services that showed lower RTE on average: 13, 18 and 21 (33.3%, Table 3); 14, 90, 96 and 124 (30,8%, Table 4); 18, 22, 27, 58, 70, 97, 110 and 115 (23.5%, Table 5); and, finally, 1, 7, 36, 56, 87, 109, 121 and 122 (26,7%, Table 6). In these services the manager´s perception on quality provided by their services surpass technical results (baseline scenario), probably because there are non-evaluated variables or circumstances in the care provision. 

On the other hand, quality variables decrease service performance only in supported accommodation services 15, 31, 32, 86 and 98 (14.7%, Table 5). This behaviour can be considered strange because here the manager´s perception on quality provided by their services underestimates their own technical results. Again, there will be additional non-assessed variables or circumstances that can affect the manager´s way of processing information to conclude in detecting frameworks of improving quality of care.

Services where the inclusion of quality variables results in a neutral behaviour in their performance are the majority, respectively: 66.7%, 69,2%, 61,8% and 73,3%. In these cases, the manager´s perception on quality provided by their corresponding services matches with their technical results. They obtain a quality according to their resources and outcomes. 

Additionally, the “Discussion” section has been enriched including the following sentences:

A recent national survey of MH supported accommodation services across England found that the quality of care was higher in MH supported housing than in residential care or floating outreach services (4). Taking into account that quality variables have not included in any RTE assessment of MH services and ecosystems (6,13,15,18,43), the question to answer here is if the perception (given by the corresponding managers) about the quality provided by the MH service is aligned or not with its technical performance. If the impact on RTE scores is negative, then the perception of the quality provided is not aligned with the technical performance (baseline RTE). In this situation, technical results do not achieve an, at least, equivalent quality, according to the manager´s opinion. If the impact can be considered neutral, then technical results and quality perception are balanced. Finally, if it is positive, managers know that the quality proved by their services is better than the corresponding neutral, probably they manage other variables that could not gathered in the QuEST study.

… Quality variables increased (sometimes not significantly) RTE on average in a relevant number of the selected services [residential houses], but the majority showed a neutral-positive profile. Quality indicators included non-technical but real service characteristics estimated by the respective managers that, in the end, improved service performance scores.

… In this type of care [supported houses] a relatively relevant number of services decrease performance scores when quality variables are included. This situation probably indicates that managers take into consideration other variables or service characteristics that end into a poorer quality results than they were expected considering the corresponding resource level and outcome production.

… As happens in residential houses, a relevant number of floating outreach services increased their performance scores significantly. Again, this increase is not aligned with the service management (resources and outcomes) showed in the baseline scenario.

Also in the “Conclusion” section 

Services that have a significant increase or decrease of their performance scores when quality variables are included into the analysis, should be studied. The observed differences to the baseline scenario (only technical) must be a consequence of something related to specific structural o managerial characteristics. These characteristics are a crucial source of information to design new interventions, policies or strategies in order to improve MH care.

---

## [Decision Letter · Decision Letter 1]

6 Sep 2021

PONE-D-21-17723R1

The critical factor: the role of quality in the performance of supported accommodation services for complex mental illness in England.

PLOS ONE

Dear Dr. Almeda,

Thank you for submitting your manuscript to PLOS ONE. After careful consideration, we feel that it has merit but does not fully meet PLOS ONE’s publication criteria as it currently stands. Therefore, we invite you to submit a revised version of the manuscript that addresses the points raised during the review process.

We look forward to receiving your revised manuscript.

Kind regards,

Dragan Pamucar

Academic Editor

PLOS ONE

Reviewers' comments:

Reviewer's Responses to Questions

**Comments to the Author**

1. If the authors have adequately addressed your comments raised in a previous round of review and you feel that this manuscript is now acceptable for publication, you may indicate that here to bypass the “Comments to the Author” section, enter your conflict of interest statement in the “Confidential to Editor” section, and submit your "Accept" recommendation.

Reviewer #1: All comments have been addressed

Reviewer #2: (No Response)

2. Is the manuscript technically sound, and do the data support the conclusions?

Reviewer #1: Yes

Reviewer #2: Partly

3. Has the statistical analysis been performed appropriately and rigorously? 

Reviewer #1: Yes

Reviewer #2: Yes

4. Have the authors made all data underlying the findings in their manuscript fully available?

Reviewer #1: Yes

Reviewer #2: No

5. Is the manuscript presented in an intelligible fashion and written in standard English?

Reviewer #1: Yes

Reviewer #2: Yes

6. Review Comments to the Author

Reviewer #1: I didn't review the paper in the previous round, but I think the reviewer has pointed the major issues that were existing in the manuscript. In my opinion, the authors have solved issues properly. Based on that I think the paper deserves to be published in a present form.

Reviewer #2: The paper “The critical factor: the role of quality in the performance of supported accommodation services for complex mental illness in England” tries to define critical quality factor for better efficiency evaluation and set up decision support system help in resource allocation.

My comments and concerns regarding the paper are given as follows.

Major comments:

1. The authors use DEA as a method of measuring RTE of supported accommodation services as part of health care system by employing commonly used input/outputs with addition of quality variables. However, the procedure and purpose of using all three methods is not clear and justified enough. Especially, it is not clear why and how authors use randomization of the original data (understanding the environmental and structural uncertainty and randomness). Is it objective to multiply number of DMU and why? How many DMUs are in the final dataset that has been used in DEA assessment?

2. There is no theoretical fundaments of used methods (DEA, Fuzzy interference and Monte Carlo) given.

Minor comments:

3. The paper is not flawlessly written. It is a bit difficult to follow the line and purpose of paper.

4. It is not necessary to have words Introduction, Objectives, Method, Results and Conclusion in the abstract. It is clear from the text what the purpose of those parts in abstract are.

5. In the Introduction, it says „Relative Technical Efficiency (RTE) is a decision support technique …”. This statement is not correct since Relative Technical Efficiency is a measure not technique.

6. “DEA can be included in a Monte Carlo simulation engine to include uncertainty and randomness in data values (all of them are considered statistical distributions) and design more realistic models (5).” This statement is not clear. Please explain how the DEA is included in a Monte Carlo Simulation engine (randomizing the data values or RTE).

7. The meaning of the following sentence after Table 6 is not clear: “This is especially relevant in some services that showed lower RTE on average: 13, 18 and 21 (33.3%, Table 3)”; What is 33.3% referred on? There are similar examples in the whole section.

8. There is no straight-forward fuzzy inference interpretation and it importance explained in discussion or conclusion.

7. PLOS authors have the option to publish the peer review history of their article (what does this mean?). If published, this will include your full peer review and any attached files.

Reviewer #1: No

Reviewer #2: No

---

## [Author Response · Author response to Decision Letter 1]

4 Oct 2021

Dear Editor and Reviewers, many thanks for your feedback, we have carried out all the changes proposed, which are highlighted in yellow in the “Revised Manuscript with Track Changes”. In this document we have developed a rebuttal letter that responds to each point raised.

Data availability

We have uploaded the dataset to Dryad Digital Repository, and it will be available for the public once the article has been published. In addition, Dryad has provided a temporal link for reviewers so they can access to the dataset. This is the link: https://datadryad.org/stash/share/1eXNj4HhRp0ztCLfgzkwO53k7Rh8q0MvVeZh_A14-xs

Nevertheless, in the text, we should maintain the original DOI for public access because the reviewers link is temporal and only for revision process. In the text: 

The dataset is available at the Dryad digital repository (https://doi.org/10.5061/dryad.j0zpc86dz). (Methods, “Setting” section).

Reviewer #1: I didn't review the paper in the previous round, but I think the reviewer has pointed the major issues that were existing in the manuscript. In my opinion, the authors have solved issues properly. Based on that I think the paper deserves to be published in a present form.

Many thanks for reviewing the manuscript and for your feedback.

Reviewer #2: The paper “The critical factor: the role of quality in the performance of supported accommodation services for complex mental illness in England” tries to define critical quality factor for better efficiency evaluation and set up decision support system help in resource allocation.

My comments and concerns regarding the paper are given as follows.

Major comments:

1. The authors use DEA as a method of measuring RTE of supported accommodation services as part of health care system by employing commonly used input/outputs with addition of quality variables. However, the procedure and purpose of using all three methods is not clear and justified enough. Especially, it is not clear why and how authors use randomization of the original data (understanding the environmental and structural uncertainty and randomness). Is it objective to multiply number of DMU and why? How many DMUs are in the final dataset that has been used in DEA assessment?

Thank you very much for your comments. We have answered your questions separately and propose the corresponding changes in the paper (in yellow).

• Relative Technical Efficiency (RTE) is a common and robust indicator for assessing the performance of any kind of comparable decision making units (in our case, mental health services). RTE is always concerned by the balance between the inputs consumed and the outputs produced, obviously looking for the best one. In order to clarify this concept we modified the introduction:

The standard performance assessment of any kind of comparable service includes studying their input consumption and output production (directly related to the inputs consumed). These inputs/outputs are mainly technical, and in the end, the best service performance is always related to the most appropriate balance between the available inputs and the produced outputs. Researchers and decision-makers can seek to reduce (minimize) the amount of inputs for a given amount of outputs (input orientation) or vice versa to increase (maximize) the output production for a specific amount of inputs (output orientation).

• In order to assess RTE we have used a non-parametric set of robust techniques that has been widely used in research and management: Data Envelopment Analysis (DEA) following the input orientation with variable return to scale as it was described in section “Decision Support System”. This technique has three main drawbacks: first, if the number of original observations (services) is not big enough DEA could not be discriminative enough (all the services would be efficient), second, it cannot understand uncertainty and randomness and, finally, it is completely blind and raw data values have to be interpreted according to a expert perspective in order to avoid wrong results. These drawbacks have to be solved when the EDeS-MH (Efficient Decision Support-Mental Health) decision support system was designed (see references number 19 and 21 for more detailed technical explanations).

• When the number of observations are less than the multiplication of the number of inputs by the number of outputs by 2 (2xNºinputsxNºoutputs), DEA can be no discriminative enough. Here the baseline scenarios have 3 inputs and 3 outputs so we need at least 18 services or more. When quality variables are included, we need, at least, 2x3x4=24 services. Surveyed residential services (9 move on services and 13 non-move on services) are not enough to carry out DEA in an standard way. On the other hand, the number of services 34 supported housing and 30 floating outreach ones can be considered likely scarce. Taking into account that all raw data should be considered under uncertainty (for example, if a specific service has 13 places it doesn’t mean that sometimes, due to more or less expected or unexpected reasons, it can have 12 or 14 or …) this is especially true when population, budged, etc. variables are included in the analysis (directly or for calculating rates). 

Uncertainty plays a relevant decisional role and it was managed by the Monte-Carlo simulation engine (more technical explanations can be seen in references 18, 19, 21 and 32). In order to carry out this statistical technique, we have to transform raw data in statistical distributions (one for each data value). We have selected standard triangular ones with small variations at the left and right hands of the modal value that was the original data value (see section “Randomization …”). When the simulation process is run, the number of observations is multiplied (Nºsimulations by Nºobservations, for example: 500 simulations by 9 move on residential services = 4,500 observations to be analysed). Simulation makes an automatic sensitivity analysis by varying original values at random according to the selected statistical distribution. This process solves the two first drawbacks (the third one is explained in the following section in this document). In order to explain this process better we have included the following sentences in the paper:

Nine move-on residential services, 13 non-move-on residential services, 34 supported housing services and 30 floating outreach services were finally analysed because they have complete quality datasets. Five hundred simulations were run by the DSS. (“Basic statistics” section)

The simulation engine was developed to address the uncertainty (data imprecision and vagueness) and randomness (unexpected facts) of real environments and to artificially multiply the number of observations (29). The inner uncertainty of any ecosystem can be overcome by transforming original data values into statistical distributions (from standard datasets to statistical distribution bases). In each simulation, the Monte Carlo simulation engine analyses a new dataset selected at random. The statistical analysis of the final results (the process is stopped when the statistical error is lower than 2.5% for the mean) includes a sensitivity analysis of the ecosystem under study. The results (RTE scores) for each DMU and scenario are statistical distributions that can be studied in a more (basic statistics) or less (stability and entropy) standard manner (19,21). The characteristics of these statistical distributions represent the potential reaction of the DMU to data changes.(“Decision Support System” section)

… Second, quality variables must then be included as outputs to investigate the influence of quality on performance. Finally, when the number of observations is low (here, the number of move-on residential services is especially low), DEA cannot be sufficiently discriminative (in the end, the methodology tends to show that all DMUs are efficient). However, as explained before, the uncertainty analysis (Monte Carlo simulation engine) multiplies the number of observations by the number of selected simulations, which overcomes this DEA drawback. (“Decision Support System” section)

2. There is no theoretical fundaments of used methods (DEA, Fuzzy interference and Monte Carlo) given.

Sorry and thank you very much for this comment that helps us to improve the paper. Again We have answered your questions separately and propose the corresponding changes in the paper.

• (Same explanation of the previous section). Relative Technical Efficiency (RTE) is a common and robust indicator for assessing the performance of any kind of comparable decision making units (in our case, mental health services). RTE is always concerned by the balance between the inputs consumed and the outputs produced, obviously looking for the best one. In order to clarify this concept we modified the introduction:

The standard performance assessment of any kind of comparable service includes studying their input consumption and output production (directly related to the inputs consumed). These inputs/outputs are mainly technical, and in the end, the best service performance is always related to the most appropriate balance between the available inputs and the produced outputs. Researchers and decision-makers can seek to reduce (minimize) the amount of inputs for a given amount of outputs (input orientation) or vice versa to increase (maximize) the output production for a specific amount of inputs (output orientation).

• Data Envelopment Analysis (DEA) is a set of well known non-parametric techniques to assess RTE (se references 19 and 21 for more technical details). In this paper we used input-oriented DEA with variable returns to scale. In order to clarify these aspects we have included the following sentences in the paper:

… In this research, the variable returns to scale DEA (31) was selected because when studying MH services, real output variations cannot be considered proportional to the corresponding input modifications (32), and constant returns to scale would involve a constant variation that cannot be considered realistic. The input-oriented DEA model was applied to assess whether service input consumption can be reduced while assuming a constant output level (9), which is especially relevant for decision-makers who must allocate finite resources to meet population needs. Output maximization (output-oriented DEA) is especially difficult and sometimes not recommendable (for example, when the system artificially tries to maximize the number of users who are moved to a service with greater independence, this can be mathematically correct but from a health care perspective it has no sense at all) when supported accommodation services are assessed. (“Decision Support System” section).

• Monte-Carlo simulation was used to include uncertainty into the analysis. This procedure allows us to develop a sensitivity analysis on the variables and scenarios. In the end, results are always statistical distributions where probabilities (to be efficient, to have an efficiency greater than 0.75, stability, entropy, etc.) are fundamental. In order to clarify this, we have included the following sentences into the paper.

…under study. The results (RTE scores) for each DMU and scenario are statistical distributions that can be studied in a more (basic statistics) or less (stability and entropy) standard manner (19,21). The characteristics of these statistical distributions represent the potential reaction of the DMU to data changes. (“Decision Support System” section).

• The fuzzy inference engine has been developed to interpret variable values according to a specific paradigm. In this paper the Balanced of Care model has been selected as this paradigm. All the variable values from the Monte-Carlo simulation engine are interpreted in terms of adequacy. For example, a low annual budget per place/bed is considered bad (not adequate) but too much budget per place/bed is also bad (not adequate), here the most appropriate values are located in between a range that is defined by the experts. Another example, a low quality score is bad and, in this case, the greater the value the greater the adequacy (greater values are always good). The fuzzy inference engine includes a knowledge-base formalized by standard IF … THEN rules. For example, IF the variable value of X is greater than a specific values THEN the value has to be transformed according to a mathematical equation or a fuzzy operator (product-sum gravity method). Our fuzzy inference engine transforms original data values (from the Monte-Carlo engine) into “interpreted” values according the information given by the experts (more technical details can be found in references 19 and 21). Trying to clarify this, we have included the following sentences into the paper.

…when inside the range. The specific references for data value interpretation have been defined by a panel of experts according to the paradigm selected and their expertise. This process followed the EbCA model (5), where an iterative sequence of expert-based reviews culminates in a consensus. Once the references for interpreting variable values are defined, the EDeS-MH automatically runs a mathematical transformation based on an equation (linear monotone transformation) or a fuzzy operator (product-sum gravity method) to obtain the “transformed” value (21). These transformed values will be analysed by DEA to determine the corresponding RTE scores (statistical distributions). (“Decision Support System” section).

Minor comments:

3. The paper is not flawlessly written. It is a bit difficult to follow the line and purpose of paper.

Thanks again for your comment. We have modify the aim of the paper in order to clarify the purpose of our research.

We include these sentences in the introduction: 

• … investment by providing relevant information for service managers to inform practice and service planning. Accordingly, this paper first presents a description of the ecosystem under study (a representative sample of supported accommodation services in England). Then, the selected variables are described and grouped into scenarios to highlight different perspectives of the ecosystem situation. Finally, the methodology used to assess ecosystem performance (including quality domains) is briefly described.

The “Procedure” section has been completely rewritten in order to improve the readability of the paper.

4. It is not necessary to have words Introduction, Objectives, Method, Results and Conclusion in the abstract. It is clear from the text what the purpose of those parts in abstract are.

Many thanks, we have deleted these words in the abstract. 

5. In the Introduction, it says „Relative Technical Efficiency (RTE) is a decision support technique …”. This statement is not correct since Relative Technical Efficiency is a measure not technique.

Many thanks, we have corrected it.

6. “DEA can be included in a Monte Carlo simulation engine to include uncertainty and randomness in data values (all of them are considered statistical distributions) and design more realistic models (5).” This statement is not clear. Please explain how the DEA is included in a Monte Carlo Simulation engine (randomizing the data values or RTE).

Thanks again for your comment. In almost any ecosystem raw data should be considered under uncertainty (for example, if a specific service has 13 places it doesn’t mean that sometimes, due to more or less expected or unexpected reasons, it can have 12 or 14 or …) this is especially true when population, budged, etc. variables are included in the analysis (directly or for calculating rates). Variable values were also obtained in a specific time and, if a wider perspective of the situation is required, researchers have to include some variability in order to be relatively sure (a probability) that the analysis can match (more or less) to a more actual situation. In order to clarify this, we have included the following into the paper:

Original data were randomized using symmetric triangular statistical distributions (5% variation on each side of the corresponding original value). This range includes feasible data variations (imprecision and vagueness). No critical stress on the ecosystem was included in the analysis (randomness). This procedure includes both data variations corresponding to ecosystem evolution (population, user mobility, etc.) and the effect of the time. (“Procedure” section).

7. The meaning of the following sentence after Table 6 is not clear: “This is especially relevant in some services that showed lower RTE on average: 13, 18 and 21 (33.3%, Table 3)”; What is 33.3% referred on? There are similar examples in the whole section.

You are completely right, sorry again. These percentages are calculated on the number of surveyed services in each type. The paragraphs have been modify accordingly.

Generally, the inclusion of quality domains highlights a neutral-positive or positive impact on the performance of supported accommodation services in England, which is especially relevant in some services that showed a lower average RTE: 13, 18 and 21 (33.3% of the surveyed services, Table 3); 14, 90, 96 and 124 (30.8% of the surveyed services, Table 4); 18, 22, 27, 58, 70, 97, 110 and 115 (23.5% of the surveyed services, Table 5); and, finally, 1, 7, 36, 56, 87, 109, 121 and 122 (26.7% of the surveyed services, Table 6). In these services, the manager´s perception of quality provided by their services surpasses technical results (baseline scenario), probably because of non-evaluated variables or circumstances in care provision.

On the other hand, quality variables decrease service performance only in supported accommodation services 15, 31, 32, 86 and 98 (14.7% of the surveyed services, Table 5).

… Services where the inclusion of quality variables results in a neutral behaviour of their performance constituted the majority, respectively: 66.7%, 69.2%, 61.8% and 73.3% of the surveyed services. In these cases, the manager´s perception of quality provided by their corresponding services matches their technical results; they obtain a quality according to their resources and outcomes.

8. There is no straight-forward fuzzy inference interpretation and it importance explained in discussion or conclusion.

Thank you very much. We have added the following sentences in the “Discussion” section.

… accommodation services (34). The knowledge base is the core of the fuzzy inference engine because it provides expert-based information for transforming the original data values (from the Monte Carlo simulation engine) according to the balanced care model. This transformation is based on experts’ opinions on the adequacy of each data value and represents a specific framework that must change throughout the time span and depend on the socioeconomic environment. Therefore, the same dataset (raw data) can result in different results according to experts’ perceptions of reality. The fuzzy inference engine can “understand” the nuances that decision-makers assume in system management.

---

## [Decision Letter · Decision Letter 2]

20 Jan 2022

PONE-D-21-17723R2The critical factor: the role of quality in the performance of supported accommodation services for complex mental illness in England.PLOS ONE

Dear Dr. Almeda,

Thank you for submitting your manuscript to PLOS ONE. After careful consideration, we feel that it has merit but does not fully meet PLOS ONE’s publication criteria as it currently stands. Therefore, we invite you to submit a revised version of the manuscript that addresses the points raised during the review process. Please submit your revised manuscript by Mar 06 2022 11:59PM. If you will need more time than this to complete your revisions, please reply to this message or contact the journal office at plosone@plos.org. Please include the following items when submitting your revised manuscript:A rebuttal letter that responds to each point raised by the academic editor and reviewer(s). You should upload this letter as a separate file labeled 'Response to Reviewers'.A marked-up copy of your manuscript that highlights changes made to the original version. You should upload this as a separate file labeled 'Revised Manuscript with Track Changes'.An unmarked version of your revised paper without tracked changes. You should upload this as a separate file labeled 'Manuscript'.If applicable, we recommend that you deposit your laboratory protocols in protocols.io to enhance the reproducibility of your results. Protocols.io assigns your protocol its own identifier (DOI) so that it can be cited independently in the future. For instructions see: https://journals.plos.org/plosone/s/submission-guidelines#loc-laboratory-protocols. Additionally, PLOS ONE offers an option for publishing peer-reviewed Lab Protocol articles, which describe protocols hosted on protocols.io. Read more information on sharing protocols at https://plos.org/protocols?utm_medium=editorial-email&utm_source=authorletters&utm_campaign=protocols.

We look forward to receiving your revised manuscript.

Kind regards,

Dragan Pamucar

Academic Editor

PLOS ONE

Journal Requirements:

Reviewers' comments:

Reviewer's Responses to Questions

**Comments to the Author**

1. If the authors have adequately addressed your comments raised in a previous round of review and you feel that this manuscript is now acceptable for publication, you may indicate that here to bypass the “Comments to the Author” section, enter your conflict of interest statement in the “Confidential to Editor” section, and submit your "Accept" recommendation.

Reviewer #1: All comments have been addressed

Reviewer #2: All comments have been addressed

2. Is the manuscript technically sound, and do the data support the conclusions?

Reviewer #1: Yes

Reviewer #2: Yes

3. Has the statistical analysis been performed appropriately and rigorously? 

Reviewer #1: Yes

Reviewer #2: Yes

4. Have the authors made all data underlying the findings in their manuscript fully available?

Reviewer #1: Yes

Reviewer #2: Yes

5. Is the manuscript presented in an intelligible fashion and written in standard English?

Reviewer #1: Yes

Reviewer #2: Yes

6. Review Comments to the Author

Reviewer #1: The authors have addressed the point of my concern. I am happy with their corrections. Hence, I would like to recommend this manuscript to be published.

Reviewer #2: The authors addressed improved paper and addressed all the comment from the previous review round. I would like to recommend to authors to include to DEA mathematical models used in the paper.

7. PLOS authors have the option to publish the peer review history of their article (what does this mean?). If published, this will include your full peer review and any attached files.

Reviewer #1: No

Reviewer #2: No

---

## [Author Response · Author response to Decision Letter 2]

22 Feb 2022

Response to Reviewers

Dear Editor and Reviewers, many thanks for assessing this research and proposed your valuable comments. We have carried out all the changes proposed, which are highlighted in yellow in the “Revised Manuscript with Track Changes”. In this document we have developed a rebuttal letter that responds to each point raised.

Journal Requirements

Please review your reference list to ensure that it is complete and correct. If you have cited papers that have been retracted, please include the rationale for doing so in the manuscript text or remove these references and replace them with relevant current references. Any changes to the reference list should be mentioned in the rebuttal letter that accompanies your revised manuscript. If you need to cite a retracted article, indicate the article’s retracted status in the References list and also include a citation and full reference for the retraction notice.

Many thanks for the comment. We have reviewed the citations and references, and we have deleted the link (http://www.ucl.ac.uk/quest) which was in “Methods>Setting” because it was not working. In addition, we have added a relevant reference for this sentence (in Methods>Setting) (highlighted in yellow in the manuscript with track changes):

“Data for supported accommodation services from 14 nationally representative local authorities in England were collected for the QuEST study, which was funded by the National Institute of Health Research (2012-2017) (4).

Reviewers’ comments

Reviewer #1: The authors have addressed the point of my concern. I am happy with their corrections. Hence, I would like to recommend this manuscript to be published.

Dear reviewer 1, many thanks for your evaluating this research and providing your valuable comments. 

Reviewer #2: The authors addressed improved paper and addressed all the comment from the previous review round. I would like to recommend to authors to include to DEA mathematical models used in the paper.

Dear referee, the DEA model has been included in section “Methods>Decision Support System” as follows (highlighted in yellow in the manuscript with track changes)

The standard DEA model is a linear programming one which structure is detailed in the revised manuscript with track changes (highlighted in yellow), in the manuscript and in the response to reviewers document. It is not possible to detail the standard DEA model in this box because it does not admit mathematical notation. 

Best regards, 

The authors

---

## [Editor Report · Decision Letter 3]

1 Mar 2022

The critical factor: the role of quality in the performance of supported accommodation services for complex mental illness in England.

PONE-D-21-17723R3

Dear Dr. Almeda,

We’re pleased to inform you that your manuscript has been judged scientifically suitable for publication and will be formally accepted for publication once it meets all outstanding technical requirements.

Kind regards,

Dragan Pamucar

Academic Editor

PLOS ONE

Additional Editor Comments (optional):

The authors have addressed the point of my concern. I am happy with their corrections. Hence, I would like to recommend this manuscript to be published.
---

## [Editor Report · Acceptance letter]

9 Mar 2022

PONE-D-21-17723R3 

The critical factor: the role of quality in the performance of supported accommodation services for complex mental illness in England. 

Dear Dr. Almeda:

I'm pleased to inform you that your manuscript has been deemed suitable for publication in PLOS ONE. Congratulations! Your manuscript is now with our production department. 

Kind regards, 

on behalf of

Dr. Dragan Pamucar 

Academic Editor

PLOS ONE